# Antiplatelet Effect of Daphnetin Is Regulated by cPLA_2_-Mediated Thromboxane A_2_ Generation in Mice

**DOI:** 10.3390/ijms24065779

**Published:** 2023-03-17

**Authors:** Preeti Kumari Chaudhary, Sanggu Kim, Soochong Kim

**Affiliations:** College of Veterinary Medicine, Chungbuk National University, Cheongju 28644, Republic of Korea

**Keywords:** daphnetin, thromboxane A_2_, cPLA_2_, ERK, 2-MeSADP, platelet

## Abstract

Coumarin derivatives have been recognized for their antithrombotic, anti-inflammatory, and antioxidant properties, and daphnetin is one of the natural coumarin derivatives isolated from *Daphne Koreana* Nakai. Although the pharmacological value of daphnetin is well documented in diverse biological activities, its antithrombotic effect has not been studied to date. Here, we characterized the role and underlying mechanism of daphnetin in the regulation of platelet activation using murine platelets. In order to check the effect of daphnetin on platelet function, we first measured the effect of daphnetin on platelet aggregation and secretion. Collagen-induced platelet aggregation and dense granule secretion were partially inhibited by daphnetin. Interestingly, 2-MeSADP-induced secondary waves of aggregation and secretion were completely inhibited by daphnetin. It is known that 2-MeSADP-induced secretion and the resultant secondary wave of aggregation are mediated by the positive feedback effect of thromboxane A_2_ (TxA_2_) generation, suggesting the important role of daphnetin on TxA_2_ generation in platelets. Consistently, daphnetin did not affect the 2-MeSADP-induced platelet aggregation in aspirinated platelets where the contribution of TxA_2_ generation was blocked. Additionally, platelet aggregation and secretion induced by a low concentration of thrombin, which is affected by the positive feedback effect of TxA_2_ generation, were partially inhibited in the presence of daphnetin. Importantly, 2-MeSADP- and thrombin-induced TxA_2_ generation was significantly inhibited in the presence of daphnetin, confirming the role of daphnetin on TxA_2_ generation. Finally, daphnetin significantly inhibited 2-MeSADP-induced cytosolic phospholipase A_2_ (cPLA_2_) and ERK phosphorylation in non-aspirinated platelets. Only cPLA_2_ phosphorylation, not ERK phosphorylation, was significantly inhibited by daphnetin in aspirinated platelets. In conclusion, daphnetin plays a critical role in platelet function by inhibiting TxA_2_ generation through the regulation of cPLA_2_ phosphorylation.

## 1. Introduction

Arterial thrombosis can cause a number of cardiovascular diseases (CVDs), including myocardial infarction, atherosclerosis, and ischemic stroke. Since platelet activation plays a substantial role in these disorders, it is clear that inhibiting platelet activation is an effective therapeutic target for arterial thrombosis. After an injury exposes the vascular sub-endothelial extracellular matrix, collagen activates platelets via the glycoprotein VI (GPVI)-mediated phospholipase Cγ2 (PLCγ2) signaling pathway by activating immuno-receptor tyrosine-based activation motif (ITAM)-containing tyrosine phosphorylation of the Fc receptor (FcR) γ-chain [1]. The activation then results in the release of various G-protein-coupled receptor (GPCR) agonists, including adenosine diphosphate (ADP), thrombin, epinephrine, and serotonin, as well as TxA_2_ generation, which then cross-talk via their respective GPCR-specific signaling pathways to cause platelet aggregation. ADP is an important agonist that induces platelet aggregation by co-activation of both the G_q_-coupled P2Y_1_ and Gi-coupled P2Y_12_ receptors [2]. Similarly, thrombin mediates its effect via protease-activated receptors (PARs) that couple to G_q_ and G_12/13_. PARs indirectly stimulate G_i_ through secreted ADP-induced P2Y_12_ receptor activation [3]. Importantly, ADP causes the generation of TxA_2_ from platelets that acts as a positive feedback mediator in activating platelet secretion and the resultant secondary wave of aggregation, and it stabilizes the hemostatic plug [4,5]. Likewise, thrombin also induces TxA_2_ generation in platelets, which contributes to potentiation of platelet aggregation and secretion induced by a low concentration of thrombin [6,7]. Platelet agonists can also activate cPLA_2_ that liberates arachidonic acid (AA) from the membrane that produces TxA_2_ [8].

Dietary supplements have a crucial role in the prevention of many human diseases, including CVDs. A number of phytochemicals have the ability to prevent and cure a number of illnesses with little to no negative side effects. Among them, coumarins, belonging to the benzopyrone family, have been used to prevent and treat a variety of diseases [9]. Some naturally occurring coumarin derivatives include warfarin, umbelliferone, esculetin, herniarin, psoralen, and imperatorin, among which some derivatives inhibit the formation of blood clots, deep vein thrombosis, and pulmonary embolism, and they are used as oral anticoagulants. However, it should be noted that the dicoumarol group of anticoagulant medications, including warfarin, is responsible for their negative effects, namely bleeding disorders. Several other naturally occurring coumarin derivatives have been shown to have an antithrombotic effect by directly inhibiting platelet activation rather than affecting the whole coagulation system. The coumarin derivative AD6 has been shown to inhibit the release of AA by inhibiting PLA_2_ in human platelets stimulated by thrombin [10]. Very recently, esculetin was shown to exhibit an antiplatelet effect by obstructing the collagen-induced PLCγ2/protein kinase C (PKC) cascade and AKT activation [11]. Similarly, another coumarin derivative, columbianadin, has been shown to inhibit PLCγ2, PKC, AKT, and ERK1/2 and markedly hinder integrin α_IIb_β_3_ activation in collagen-activated platelets [12]. Esculetin has also been shown to inhibit nuclear factor kappa B (NF-κB)-dependent α_IIb_β_3_ activation [13].

Daphnetin is a natural coumarin derivative extracted from plants of the genus *Daphne* [14]. It has been demonstrated to have a number of biological characteristics, including anti-inflammatory, anti-hypoxic, antimicrobial, anti-proliferative, and anti-cancerous [15,16,17]. In China, coagulation disease is now being treated therapeutically with daphnetin, suggesting that it can be a promising therapeutic approach for thrombosis. Although the pharmacological value of daphnetin is well-documented in diverse biological activities, its potential to interfere with platelet signaling and the molecular mechanism behind it are two aspects of its antithrombotic impact that have not yet been investigated.

In the present study, we examined the effect of daphnetin on platelet function and explored the molecular mechanism using murine platelets. We have shown that daphnetin inhibits 2-MeSADP-induced secretion and the resultant secondary wave of aggregation in non-aspirinated platelets. We have further shown that daphnetin does not have any effect on 2-MeSADP-induced aggregation in aspirinated platelets, suggesting that daphnetin exerts its antiplatelet effect by regulating TxA_2_ generation. Finally, we have shown that 2-MeSADP-induced TxA_2_ generation and cPLA_2_ phosphorylation are inhibited in the presence of daphnetin. In conclusion, daphnetin plays a role in platelet function by inhibiting TxA_2_ generation through the regulation of cPLA_2_ phosphorylation.

## 2. Results

### 2.1. The Effect of Daphnetin on GPVI-Mediated Platelet Activation

To check whether daphnetin, like other coumarin derivatives, has any role in the regulation of GPVI-mediated platelet response, we first stimulated the platelets with various concentrations of collagen and measured the platelet aggregations and dense granule secretion. As shown in Figure 1, daphnetin significantly inhibited collagen-induced dense granule secretion in a dose-dependent manner while only a high concentration of daphnetin inhibited a low concentration of collagen-induced platelet aggregation. The data suggest that daphnetin has a minor role in GPVI-mediated platelet response, unlike other coumarin derivatives.

### 2.2. Daphnetin Regulates 2-MeSADP-Induced Secondary Wave of Aggregation and Secretion in Platelets

To evaluate the effect of daphnetin on GPCR-mediated platelet function, we checked the effect of daphnetin on 2-MeSADP-induced platelet aggregation and secretion. In Figure 2, the secondary waves of platelet aggregation and secretion induced by both low and high concentrations of 2-MeSADP were completely inhibited by daphnetin.

It is well established that 2-MeSADP-induced secretion and the resultant secondary wave of platelet aggregation are dependent on TxA_2_ generation [4,18]. Therefore, we pretreated the platelets with aspirin to block the positive feedback effect of generated TxA_2_ and measured the platelet aggregation and dense granule secretion in response to 2-MeSADP in the presence of daphnetin. Interestingly, daphnetin did not further block 2-MeSADP-induced platelet aggregation in aspirin-treated platelets (Figure 3), suggesting the effect of daphnetin on TxA_2_ generation.

### 2.3. Daphnetin Regulates Platelet Aggregation and Secretion Induced by Low Concentration of Thrombin

To evaluate the effect of daphnetin on other agonist-induced platelet aggregation and secretion, we stimulated the platelets with thrombin in the presence of daphnetin. As shown in Figure 4, daphnetin inhibited a low concentration of thrombin-induced platelet aggregation and dense granule secretion but did not affect aggregation or secretion when platelets were stimulated with high concentrations of thrombin.

### 2.4. Daphnetin Regulates 2-MeSADP- and Thrombin-Induced TxA_2_ Generation in Platelets

To confirm the effect of daphnetin on TxA_2_ generation, we stimulated the platelets with 2-MeSADP and thrombin and measured the total amount of TxA_2_ generation. We found that 2-MeSADP- and thrombin-induced TxA_2_ generation was significantly inhibited by daphnetin in a dose-dependent manner (Figure 5), confirming the important role of daphnetin on TxA_2_ generation.

### 2.5. Daphnetin Inhibits 2-MeSADP-Induced cPLA_2_ and ERK Phosphorylation in Platelets

To determine the molecular mechanism involved in the regulation of TxA_2_ generation by daphnetin, we stimulated platelets with 2-MeSADP in the presence and absence of aspirin and measured the phosphorylation of cPLA_2_, ERK, and AKT. As shown in Figure 6, 2-MeSADP-induced cPLA_2_ phosphorylation was completely inhibited in the presence of daphnetin in a dose-dependent manner in both non-aspirinated and aspirinated platelets. However, 2-MeSADP-induced ERK phosphorylation was significantly inhibited by daphnetin only in non-aspirinated platelets, while daphnetin showed no effect on 2-MeSADP-induced ERK phosphorylation in aspirinated platelets. In addition, daphnetin did not have any effect on 2-MeSADP-induced AKT phosphorylation in both non-aspirinated and aspirinated platelets. The data indicate that daphnetin regulates TxA_2_ generation through direct inhibition of cPLA_2_ phosphorylation in platelets.

## 3. Discussion

A platelet plays a vital role in thrombosis and hemostasis. The antiplatelet medications that are most frequently prescribed include cyclooxygenase inhibitors, ADP receptor antagonists, and GPIIb/IIIa receptor antagonists [19]. Numerous phytochemicals have the preventive and therapeutic capacity to effectively treat a variety of diseases with little or no adverse effects. Because of their potential to prevent and treat a number of diseases, coumarins, among others, have sparked a revolution in the research community and have also been used as anticoagulants recently [8]. The pharmacological value of daphnetin is well documented in diverse biological activities but its antithrombotic effect is not well established to date. Therefore, our objective was to investigate the role of daphnetin and its underlying mechanism on the regulation of platelet activation using murine platelets.

Coumarin derivatives, such as esculetin and columbianadin, have been known to exert their antiplatelet effect by targeting PLCγ2 that regulates PKC-mediated MAP kinase activation, TxA_2_ generation, and granule secretion downstream of GPVI-mediated platelet signaling, while they have no effect on thrombin-, U46619- (a TxA_2_ analog), and AA-induced platelet aggregation [11,12]. However, daphnetin only partially inhibited collagen-induced platelet aggregation and secretion. If daphnetin inhibits PLCγ_2_ like other coumarin derivatives [11,12], we would not expect to see the aggregation in collagen-stimulated platelets in the presence of daphnetin. These data suggest that daphnetin has a minor effect in GPVI-mediated platelet response in contrast to other coumarin derivatives and may exert its antiplatelet effect via different signaling mechanisms.

Platelet GPCRs are clinically significant GPCRs that are the targets of numerous medications used to treat bleeding problems or to prevent clotting. Recently, a study that evaluated the bioactivities of many daphnetin derivatives proposed the potential of novel daphnetin-based GPCR inhibitors or activators as new therapeutic agents for the treatment of metabolic disorders [20]. This suggests that daphnetin may have a role in regulating GPCR-mediated platelet function. ADP, TxA_2_, and thrombin are major agonists that promote the development and stability of thrombi via GPCR-mediated signaling pathways in platelets. Unlike GPVI signaling, GPCR-mediated signaling activates G_q_-dependent PLCβ to facilitate PKC activation, TxA_2_ generation, and granule release and causes platelet activation. We found that dense granule secretion and the resultant secondary wave of platelet aggregation induced by both low and high concentrations of 2-MeSADP were completely inhibited by daphnetin. It has been shown that ADP-induced secretion and the resultant secondary wave of aggregation are dependent on the positive feedback effect of generated TxA_2_ [4]. The inhibition of TxA_2_ generation reduces ischemic events in clinical studies, suggesting a significant role for TxA_2_ in in vivo regulation of hemostasis and thrombosis [21,22]. Upon blockage of the contribution of TxA_2_ generation by aspirin treatment, daphnetin did not show any additional inhibitory effect on 2-MeSADP-induced platelet aggregation compared to a non-aspirin-treated platelet. Taken together, our data indicate that daphnetin affects the 2-MeSADP-induced secondary wave of aggregation and secretion through the inhibition of TxA_2_ generation in platelets.

Thrombin is a more potent agonist than ADP that causes much stronger G_q_ and G_12/13_ stimulation [23]. It has been demonstrated that thrombin-induced platelet aggregation necessitates secreted ADP only at lower concentrations and is mediated primarily through a family of G_q_-coupled PARs, independently of secreted ADP-mediated G_i_ signaling [3]. Thrombin-induced platelet aggregation is affected by the positive feedback effect of generated TxA_2_ at a lower concentration but does not depend on TxA_2_ generation and resultant secretion at a higher concentration [3,6]. Thus, unlike ADP, thrombin-induced platelet aggregation and secretion do not require TxA_2_ generation. Consistently, we found that daphnetin only affected the low concentration of thrombin-induced platelet aggregation and secretion, while the high concentration of thrombin-induced platelet aggregation and secretion were not affected by daphnetin. Importantly, we found that 2-MeSADP- and thrombin-induced TxA_2_ generation was significantly inhibited by daphnetin in a dose-dependent manner, confirming the effect of daphnetin on TxA_2_ generation. Similar to our findings, Galli et al. [10] also observed that AD6, the other coumarin derivative, was able to decrease the generation of TxB_2_ from endogenous AA in platelet-rich plasma (PRP) stimulated with collagen. Altogether, our data indicate that daphnetin affects the 2-MeSADP and thrombin-induced platelet function by inhibiting TxA_2_ generation in platelets.

The rate-limiting step in TxA_2_ biosynthesis by platelets is the activation of cPLA_2_ with the production of free AA as a substrate for cyclooxygenase (COX) [24,25]. We found that daphnetin completely inhibited 2-MeSADP-induced cPLA_2_ phosphorylation in both non-aspirinated and aspirinated platelets, demonstrating that daphnetin inhibits TxA_2_ generation by specifically targeting cPLA_2_ phosphorylation in platelets. It has been demonstrated that ADP-induced TxA_2_ generation is regulated by ERK, thus establishing an important role of ERK in TxA_2_ generation in platelets [4,5,18,26]. It has also been reported that mitogen-activated protein kinase (MAPK) phosphorylation including ERK is dependent on TxA_2_-stimulated activation of the TP receptor in various cell lines [27,28,29]. Additionally, it has been demonstrated that ERK phosphorylation is significantly intensified in a time-dependent manner upon stimulation of platelet with TxA_2_, thus indicating the significance of ERK in TxA_2_-mediated platelet activation [30]. Interestingly, 2-MeSADP-induced ERK phosphorylation was significantly inhibited by daphnetin in non-aspirinated platelets while this ERK phosphorylation was not affected by daphnetin in aspirinated platelets. It has been shown that ERK has the ability to phosphorylate cPLA_2_ on Ser505, which somewhat boosts cPLA_2_’s catalytic activity [31]. However, we found that daphnetin inhibited 2-MeSADP-induced cPLA_2_ phosphorylation in aspirinated platelets while having no effect on ERK phosphorylation, indicating that daphnetin does not directly target ERK to regulate cPLA_2_’s activity. These results indicate that daphnetin in the absence of TxA_2_ generation does not have any effect on ERK phosphorylation, suggesting that daphnetin regulates ERK phosphorylation by inhibiting TxA_2_ generation. It has been well established that AKT is activated downstream of phosphoinositide 3-kinase (PI3K) by ADP in a P2Y_12_ receptor-dependent G_i_ pathway in platelets [32], but the PI3K-AKT pathway does not play any role in the regulation of TxA_2_ generation. Daphnetin had no effect on 2-MeSADP-induced AKT phosphorylation in both non-aspirinated and aspirinated platelets. Taken together, our data demonstrate that daphnetin regulates 2-MeSADP-induced TxA_2_ generation by directly regulating cPLA_2_ phosphorylation in platelets.

In in vivo studies, inhibition of ERK has been shown to significantly increase survival rates in a pulmonary thromboembolism assay and time to arterial occlusion, highlighting a potential in vivo role for ERK in the regulation of thrombosis and hemostasis [33,34]. Likewise, TxA_2_ receptor knockout mice have been reported to have mild bleeding disorders [35]. As daphnetin regulates the platelet function by inhibiting TxA_2_ generation, we expect that daphnetin would have similar effects on platelet function in vivo as those were seen in the previous studies. TxA_2_ is a product of the oxidative metabolism of AA generated by the platelet, thereby acting in an autocrine manner to stimulate TxA_2_ prostanoid (TP) receptors [36]. Thus, the TxA_2_ pathway has been the constant target for antiplatelets. TP receptor is one such GPCR that remains resistant for therapeutic interventions. Aspirin has been frequently used as an irreversible COX inhibitor that reduces TxA_2_ generation and TP receptor stimulation. Despite the efficacy of aspirin, there is still interest in developing direct TP receptor antagonists in order to preserve the beneficial effects of other prostanoids such as gastric mucosal protection that are lost upon global inhibition of COX [36,37]. As daphnetin is a natural derivative, it would be beneficial to carry out an in vivo assay to further validate the physiological significance of daphnetin in thrombosis and hemostasis, and whether it can be used as an alternative therapeutic source of TxA_2_ inhibitors to treat various cardiovascular diseases.

In conclusion, we demonstrate that the antiplatelet effect of daphnetin is mediated by the inhibition of TxA_2_ generation through the regulation of cPLA_2_ phosphorylation. Further, it regulates generated TxA_2_-mediated ERK phosphorylation indirectly to regulate the platelet functional responses.

## 4. Materials and Methods

### 4.1. Materials

Collagen was purchased from Chrono-log Corporation (Havertown, PA, USA). The 2-MeSADP, thrombin, daphnetin, apyrase (type V), prostaglandin E_1_ (PGE_1_), sodium citrate, and acetylsalicylic acid (ASA) were bought from Sigma (St. Louis, MO, USA). Anti-phospho-cPLA_2_ (Ser505), anti-cPLA_2_, anti-phospho-ERK (Thr202/Tyr204), anti-ERK, Anti-phospho-AKT (Ser473), and anti-AKT antibodies were from Cell Signaling Technology (Beverly, MA, USA). Horseradish peroxidase-labeled secondary antibody was purchased from Santa Cruz Biotechnology (Santa Cruz, CA, USA). TxB_2_ ELISA kit was from Enzo Life Sciences (Exeter, UK). All additional chemicals were of reagent grade.

### 4.2. Animals

We obtained 8-week-old C57BL/6 mice from Orient Bio, Inc. (Seongnam-si, Gyeonggi-Do, Republic of Korea), and we housed them in a semi-pathogen-free facility. All animal experiments were performed with approval from Chungbuk National University Animal Ethics Committee (CBNUA-873-15-02).

### 4.3. Mouse Platelet Isolation

Isolation of platelets from murine blood was performed as previously described [38]. Whole blood was drawn by cardiac venipuncture from an equal number of both male and female mice; sodium citrate was added and spun at 100× *g* for 10 min at room temperature (RT) to produce PRP. The PRP was given 1 mM ASA for 30 min at 37 °C as part of the aspirin therapy. After being centrifuged at 400× *g* for 10 min, the platelets were pelleted. The concentration of platelets was adjusted to 2 × 10^8^ cells/mL by re-suspending the pellets in Tyrode’s buffer (pH 7.4) with 0.05 units/mL of apyrase.

### 4.4. Platelet Aggregation and Dense Granule Secretion

A Lumi-aggregometer (Chrono-Log, Havertown, PA, USA) was used to measure agonist-induced platelet aggregation and granule secretion as described previously [38]. Briefly, washed platelets (250 µL) were stimulated with the different agonists, and changes in light transmission were measured. By using luciferin/luciferase reagent, the adenosine triphosphate (ATP) release from platelets was monitored to evaluate platelet dense granule secretion according to the manufacturer’s instructions.

### 4.5. Measurement of TxA_2_ Generation

Non-aspirinated washed murine platelets were prepared at a concentration of 2 × 10^8^ platelets/mL. Platelets were stimulated with an agonist for 3.5 min in a Lumi-aggregometer and the reaction was stopped by snap freezing. Samples were kept at −80 °C until T_X_B_2_ levels were measured. Levels of T_X_B_2_ were measured in duplicates using a TxB_2_ ELISA Kit (Enzo, Exeter, UK), as per the manufacturer’s instructions.

### 4.6. Immuno-Blotting

Phosphorylations were quantified after activating the platelets with 2-MeSADP as described previously [39]. Platelets were activated and the reaction was stopped by the addition of 6.6 N perchloric acid. To separate platelet lysates, 10% SDS/PAGE was used before they were transferred to nitrocellulose membranes. Non-specific binding sites were blocked by incubating in Tris-buffered saline/Tween-milk protein for 30 min at RT. Anti-phospho-cPLA_2_ (Ser505), anti-cPLA_2_, anti-phospho-ERK (Thr202/Tyr204), anti-ERK, anti-phospho AKT (Ser473), or anti-AKT antibodies at dilutions of 1:2000, 1:2000, 1:2000, 1:2000, 1:5000, and 1:5000, respectively, were incubated with membranes overnight at 4 °C. Immune reactivity was detected after membranes were probed with goat anti-rabbit antibody using iBright^TM^ CL1500 imaging system (Invitrogen, Waltham, MA, USA).

### 4.7. Data Analysis

All statistical calculations were determined using Prism software (version 9.1). The data were shown as mean ± standard error (SE). Statistical significance was established using the one-way analysis of variance (ANOVA) followed by a post hoc Dunnett’s multiple comparison test. The normality test was conducted by using Shapiro–Wilk test and data were found to be normally distributed (*p* > 0.05).

## Figures and Tables

**Figure 1 ijms-24-05779-f001:**
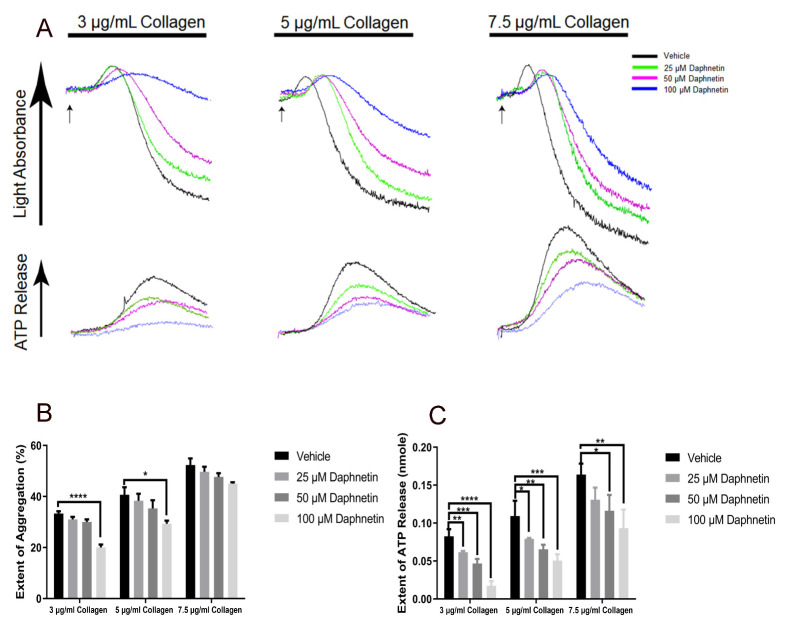
The effect of daphnetin on collagen-induced platelet aggregation and secretion. (**A**) Washed murine platelets were stimulated with GPVI agonist 3 µg/mL, 5 µg/mL, and 7.5 µg/mL collagen in the presence of different concentrations of daphnetin for 3.5 min. Platelet aggregation (top) and ATP secretion (bottom) were analyzed in a Lumi-aggregometer. Data are representative of three independent experiments (*n* = 3). Quantification of the (**B**) extent of aggregation and (**C**) dense granule secretion from panel A. Data are presented as mean ± SE *, *p* < 0.05; **, *p* < 0.01; ***, *p* < 0.005; ****, *p* < 0.0001.

**Figure 2 ijms-24-05779-f002:**
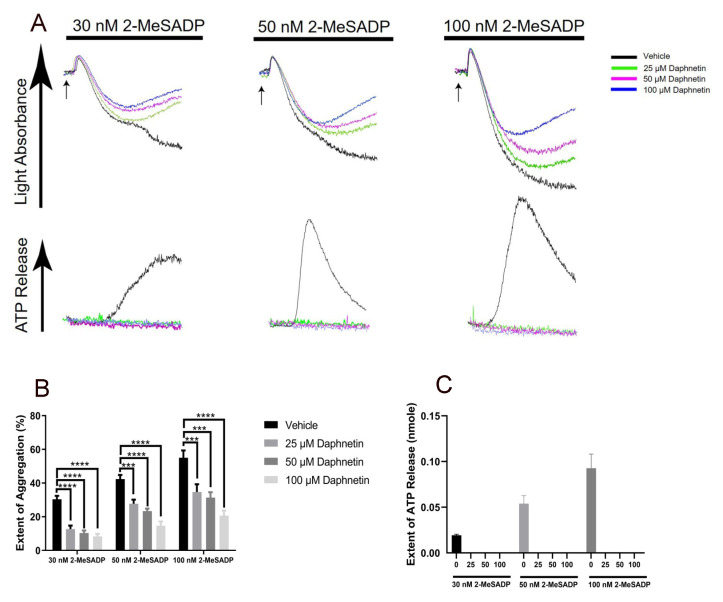
The effect of daphnetin on 2-MeSADP-induced aggregation and secretion in non-aspirinated platelets. (**A**) Washed murine platelets were stimulated with 30 nM, 50 nM, and 100 nM 2-MeSADP in the presence of different concentrations of daphnetin. All data are representative of three independent experiments (*n* = 3). Quantification of the (**B**) extent of aggregation and (**C**) dense granule secretion from panel A. Data are shown as mean ± SE ***, *p* < 0.001; ****, *p* < 0.0001.

**Figure 3 ijms-24-05779-f003:**
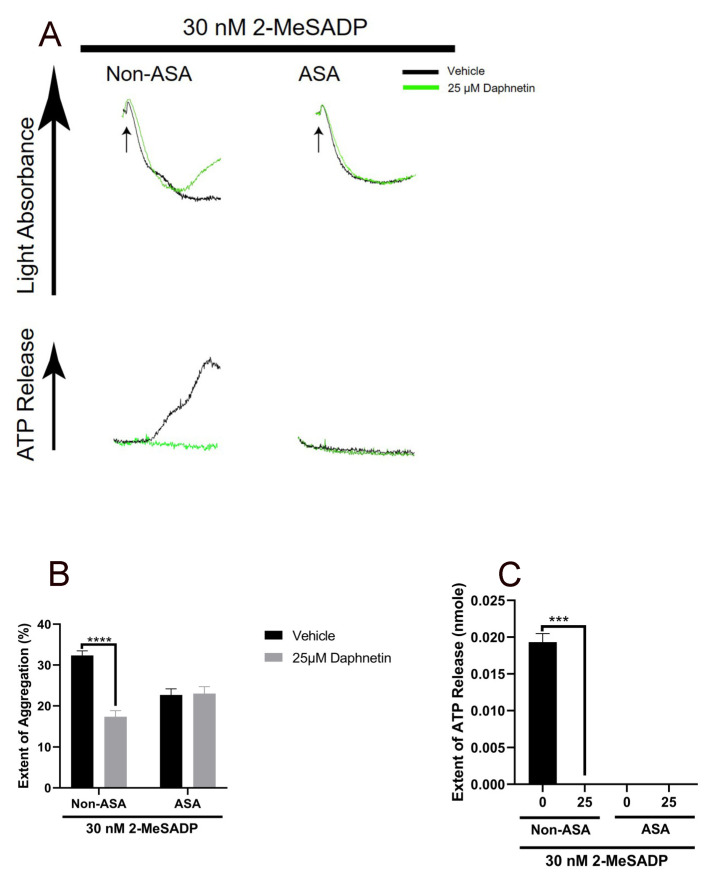
No additional effect of daphnetin on 2-MeSADP-induced platelet aggregation in aspirinated platelets. (**A**) Non-ASA- and ASA-treated murine platelets were stimulated with 30 nM 2-MeSADP in the presence of 25 µM daphnetin. All tracings shown are representative of at least three different experiments (*n* = 3). Quantification of the (**B**) extent of aggregation and (**C**) dense granule secretion from panel A. Data are presented as mean ± SE ***, *p* < 0.001; ****, *p* < 0.0001.

**Figure 4 ijms-24-05779-f004:**
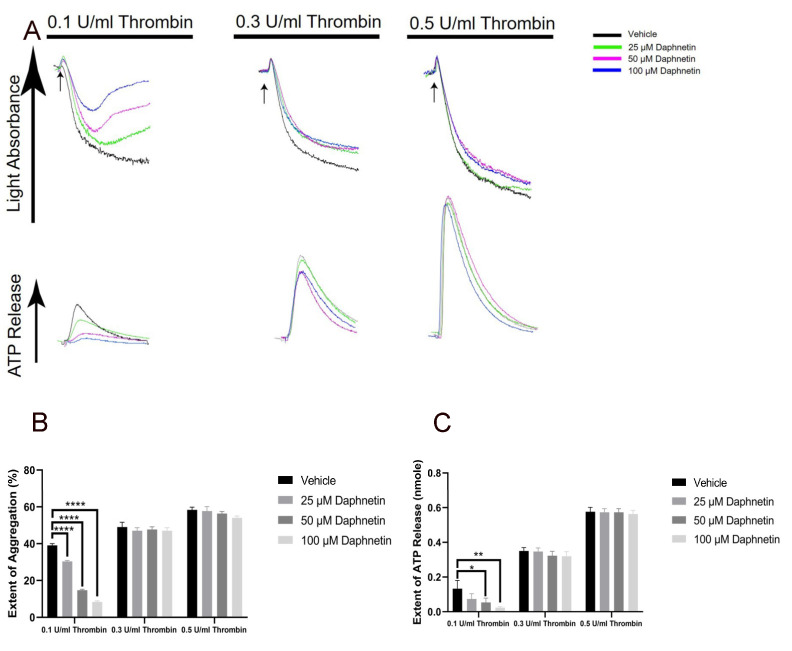
The effect of daphnetin on thrombin-induced platelet aggregation and secretion. (**A**) Washed murine platelets were stimulated with 0.1 U/mL, 0.3 U/mL, and 0.5 U/mL thrombin in the presence of different concentrations of daphnetin. Quantification of the (**B**) extent of aggregation and (**C**) dense granule secretion from panel A. Data are representative of three independent experiments (*n* = 3) and are presented as mean ± SE *, *p* < 0.05; **, *p* < 0.01; ****, *p* < 0.0001.

**Figure 5 ijms-24-05779-f005:**
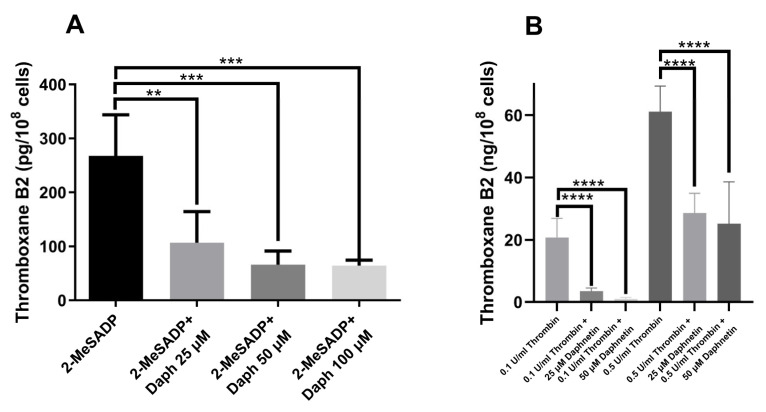
The effect of daphnetin on TxA_2_ generation induced by 2-MeSADP and thrombin in platelets. Washed murine platelets were stimulated with (**A**) 100 nM 2-MeSADP and (**B**) 0.1 U/mL and 0.5 U/mL thrombin in the presence of different concentrations of daphnetin at 37 °C for 3.5 min under stirring condition and TxB_2_ generation were measured in duplicates by ELISA assay. Data are representative of three independent experiments (*n* = 3) and are presented as mean ± SE **, *p* < 0.01; ***, *p* < 0.001; ****, *p* < 0.0001.

**Figure 6 ijms-24-05779-f006:**
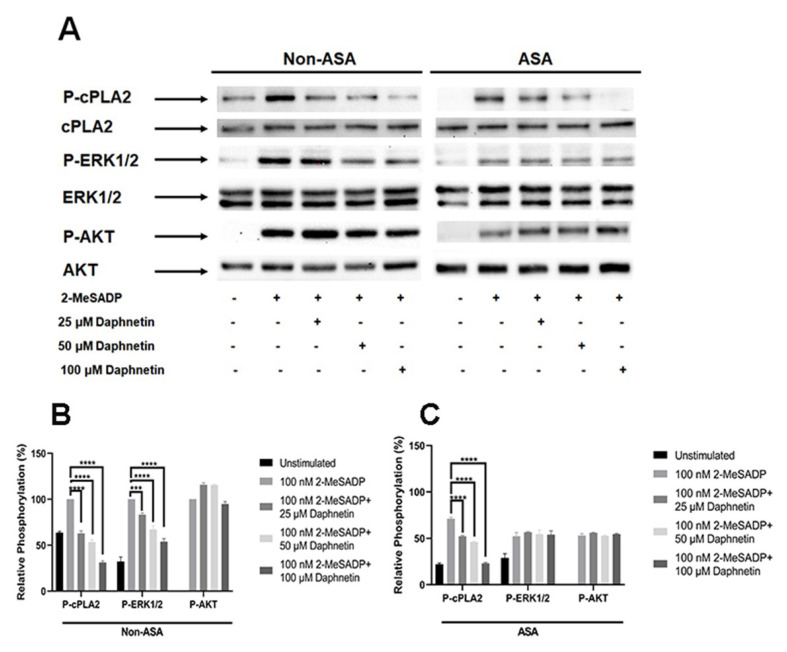
The effect of daphnetin on cPLA_2_ and ERK phosphorylation induced by 2-MeSADP in platelets. (**A**) Non-ASA- and ASA-treated washed platelets were stimulated with 100 nM 2-MeSADP in the presence of different concentrations of daphnetin at 37 °C for 2 min. The lysate volume corresponding to the same number of platelets was loaded. The membrane was probed with anti-phospho-cPLA_2_ (Ser505), anti-cPLA_2_, anti-phospho-ERK, anti-ERK, anti-phospho-AKT (Ser473), or anti-AKT antibodies. Data shown are representative of three independent experiments (*n* = 3). Blots in (**B**) non-ASA- and (**C**) ASA-treated platelets are quantified and presented as mean ± SE ***, *p* < 0.001; ****, *p* < 0.0001.

## Data Availability

Not applicable.

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
