# Peer review of "Antiplatelet Effect of Daphnetin Is Regulated by cPLA2-Mediated Thromboxane A2 Generation in Mice"

_ijms, 2023, doi:10.3390/ijms24065779_

Round 1
Reviewer 1 Report (Previous Reviewer 1)
The manuscript entitled " Anti-platelet Effect of Daphnetin is Regulated by ERK-mediated Thromboxane A2 Generation" in which the authors examined the impact of the coumarin by daphnetin on the regulation of platelet activation using murine platelets. They found that daphnetin has anti-platelet effect that is mediated by the inhibition of Thromboxane A2 generation through the regulation of ERK phosphorylation not AKT phosphorylation.
The data presented here supports the authors’ conclusions. The scientific narrative is well structured and flows naturally from one idea to the next. The findings are novel and important. Moreover, while this report provides a novel therapeutic compound for the treatment of thromboembolic disorders.
The revised manuscript is improved compared to prior revision. All my comments were adequately answered and explained by the authors. Therefore, I consider that the revised manuscript is acceptable and suitable for publication in International Journal of Molecular Sciences.

Author Response
Response to Reviewer 1
Comment: The revised manuscript is improved compared to prior revision. All my comments were adequately answered and explained by the authors. Therefore, I consider that the revised manuscript is acceptable and suitable for publication in International Journal of Molecular Sciences.
Response: We would like to thank the reviewer for the evaluation of the manuscript and helpful suggestions. We are grateful that the reviewer felt that we addressed concerns enough. We also appreciate the reviewer’s decision to recommend the work for publication in the International Journal of Molecular Sciences.
Reviewer 2 Report (New Reviewer)
This is a review of the manuscript by Chaudhary et al. entitled "Anti-platelet Effect of Daphnetin is Regulated by ERK-mediated Thromboxane A2 Generation".
This study examines the pharmacological effects of the coumarin derivative daphnetin on mouse platelet aggregation and secretion in response to a nonhydrolyzable analog of ADP, collagen and thrombin. Methods are appropriate and thorough and statistical analysis is sufficient. Overall, this is an interesting and high-quality study, but the mechanism and impact of daphnetin in platelets could be clarified further with more experiments.
The title should mention mouse platelets since no experiments were conducted in human platelets.
Concentrations of daphnetin are not physiological: 25 mM is extremely high for ex vivo experiments. Lower concentrations of daphnetin should be tested in all experiments. At lower daphnetin concentrations it may be required to lower the dose of agonist; for example, the concentration of collagen could be lowered to at least 0.5 - 1 ug/mL in Figure 1.
Related to this, inhibitory concentration (IC50) should be determined using a range of daphnetin concentrations. This would allow comparisons to other drugs and inhibitors. This may be most relevant for secretion.
In Figure 3, aspirin is used to block TXA2 production, which is commonly done. However, aspirin (ASA) can acetylate many off-target proteins so perhaps a TXA2 receptor (TP) inhibitor like terutroban could be used to see if the effects of ASA are recapitulated. This would provide a more precise inhibition of TXA2 action compared to global and possibly nonspecific inhibition with ASA. Terutroban could also be used to block ERK phosphorylation as in Figure 6 to show that 1) TXA2 activates ERK and 2) daphnetin in combination with terutroban does not have additive effects. Regardless ASA should be used for the same reasons in Figure 6.
It is not clear how ERK is being inhibited by daphnetin. One simple explanation is that daphnetin directly inhibits phospholipase C beta, which is downstream of the ADP receptor P2Y1 and thrombin receptors. Since other coumarin derivatives inhibit PLC gamma, this may be a possibility. Can the authors test PLC beta activity? The coumarin derivative AD6 inhibits phospholipase A2 (PLA2), so is it possible that this enzyme is also inhibited by daphnetin? Other molecules such as Rap1 and PKC are downstream of P2Y1 and thrombin and also activate ERK, but that may be outside the scope of the current study.
Another interesting experiment would be to use arachidonic acid (AA) as an agonist. If daphnetin is acting at the level of ERK or phospholipase A2, then AA should bypass and show normal aggregation and secretion in the presence of daphnetin. This would be additional evidence that daphnetin affects ERK, but does not rule out direct interference with PLC gamma or PLA2.
Author Response
We would like to thank the reviewer for the evaluation of the manuscript and helpful suggestions. We are thankful to the reviewers for the insightful comments and criticisms. We have clarified the issues raised by reviewers. The responses to the specific comments for the reviewers are given below:
Comment: The title should mention mouse platelets since no experiments were conducted in human platelets.
Response: We thank the reviewer for the comment. We have revised the title to “Anti-platelet Effect of Daphnetin is Regulated by cPLA2-mediated Thromboxane A2 Generation in Mice”. Please refer to the revised version of the manuscript.
Comment: Concentrations of daphnetin are not physiological: 25 mM is extremely high for ex vivo experiments. Lower concentrations of daphnetin should be tested in all experiments. At lower daphnetin concentrations it may be required to lower the dose of agonist; for example, the concentration of collagen could be lowered to at least 0.5 - 1 ug/mL in Figure 1.
Related to this, inhibitory concentration (IC50) should be determined using a range of daphnetin concentrations. This would allow comparisons to other drugs and inhibitors. This may be most relevant for secretion.
Response: We thank the reviewer for the comment. In our experiment, we have followed the standard low, intermediate, and high concentrations of various agonists to check the platelet functional responses in the presence of daphnetin. The IC50 value of daphnetin has been shown to vary from 7.65 µM to 25.01 µM depending on different targets such as EFGR, PKA, or PKC. Likewise, the EC50 value also ranges from 31 µM to more than 650 µM. Therefore, the use of daphnetin concentration is highly variable depending on cell type and time for treatment in the different in vitro experimental settings. Concentrations equivalent to 25-100 µM; 10-50 µM; 140-570 µM have been used in splenocytes proliferation assays, EGR receptor solution, and H. pylori clinical isolates experiments, respectively. As this is the first time we are using daphnetin to study its effect on platelet function, we chose to use the concentrations 25-100 µM to see the dose-dependent effect of daphnetin on platelet function.
Comment: In Figure 3, aspirin is used to block TXA2 production, which is commonly done. However, aspirin (ASA) can acetylate many off-target proteins so perhaps a TXA2 receptor (TP) inhibitor like terutroban could be used to see if the effects of ASA are recapitulated. This would provide a more precise inhibition of TXA2 action compared to global and possibly nonspecific inhibition with ASA. Terutroban could also be used to block ERK phosphorylation as in Figure 6 to show that 1) TXA2 activates ERK and 2) daphnetin in combination with terutroban does not have additive effects. Regardless ASA should be used for the same reasons in Figure 6.
Response: We thank the reviewer for the comment. As the reviewer also mentioned, we have employed the frequently used COX-inhibitor ASA to block the positive-feedback effect of TxA2 and comprehend the mechanism underlying the anti-platelet activity of daphnetin. Even though, we agree with the reviewer of using specific TxA2 receptor (TP) inhibitor for a precise inhibition of TxA2, we expect that the use of ASA or terutroban would lead to the same effect since we have confirmed the direct inhibitory effect of daphnetin on TxA2 generation in Figure 5. As per the reviewer’s suggestion, we measured the 2-MeSADP-induced ERK phosphorylation in both non-aspirinated and aspirinated platelets and found that daphnetin only inhibits 2-MeSADP-induced ERK phosphorylation in non-aspirinated platelets, indicating that TxA2 activates ERK and daphnetin does not have any additive effect in the absence of TxA2 generation. The possible mechanism involved and rationale that may be involved are discussed in the revised version of the manuscript and Figure 6 has been updated.
Comment: It is not clear how ERK is being inhibited by daphnetin. One simple explanation is that daphnetin directly inhibits phospholipase C beta, which is downstream of the ADP receptor P2Y1 and thrombin receptors. Since other coumarin derivatives inhibit PLC gamma, this may be a possibility. Can the authors test PLC beta activity? The coumarin derivative AD6 inhibits phospholipase A2 (PLA2), so is it possible that this enzyme is also inhibited by daphnetin? Other molecules such as Rap1 and PKC are downstream of P2Y1 and thrombin and also activate ERK, but that may be outside the scope of the current study.
Response: We thank the reviewer for the insightful comment. When we examined the cPLA2 phosphorylation in the revision, we discovered that daphnetin completely inhibited 2-MeSADP-induced cPLA2 phosphorylation in both non-aspirinated and aspirinated platelets in a dose-dependent manner. Since TxA2 is the major product of cPLA2 and daphnetin did not affect 2-MeSADP-induced ERK phosphorylation in aspirinated platelets, our data indicate that daphnetin inhibits the TxA2 generation by directly inhibiting cPLA2 phosphorylation and indirectly regulates TxA2 generation-mediated ERK phosphorylation to exert its anti-platelet effect.
Comment: Another interesting experiment would be to use arachidonic acid (AA) as an agonist. If daphnetin is acting at the level of ERK or phospholipase A2, then AA should bypass and show normal aggregation and secretion in the presence of daphnetin. This would be additional evidence that daphnetin affects ERK, but does not rule out direct interference with PLC gamma or PLA2.
Response: We thank the reviewer for the insightful comment. As mentioned above, we examined the effect of daphnetin on cPLA2 and confirmed the effect of daphnetin on TxA2 generation through the regulation of cPLA2 phosphorylation. Thus, we anticipate and concur with the reviewer that AA will bypass and show normal aggregation and secretion in the presence of daphnetin.
Round 2
Reviewer 2 Report (New Reviewer)
The authors propose that daphnetin is acting similarly to aspirin in that daphnetin is lowering TXA2 levels through inhibition of cPLA2. The revised manuscript is improved, but more interpretation of results is needed. An additional experiment for Figure 6 was added in which cPLA2 phosphorylation was measured using a specific antibody after treatment with daphnetin in the presence or absence of aspirin. Daphnetin reduced cPLA2 phosphorylation under both conditions, which was logically interpreted to mean that cPLA2 activity was reduced and that daphnetin was acting through cPLA2 by "directly regulating cPLA2 phosphorylation in platelets". This would explain decreased TXA2 levels and reduced ERK phosphorylation since TxA2 feeds back through an autocrine loop that leads to ADP secretion, Rap activation and ERK phosphorylation, which the authors acknowledge in references 4,5,18 and 27.
These data imply a direct inhibition of cPLA2, which makes sense except for the fact that the antibody used to detect cPLA2 phosphorylation recognizes a known phosphorylation site for ERK (Ser505 on cPLA2). Are the authors suggesting that daphnetin inhibits ERK activity directly or does daphnetin bind cPLA2 and prevent phosphorylation at this site resulting in lower catalytic activity? This is a complicated signaling pathway with multiple positive feedback loops, so more discussion of how daphnetin may be regulating an ERK phosphorylation site on cPLA2 is warranted (preferably near Line 266 in the Discussion).
On a technical level, the new P-ERK1/2 blot from ASA-treated platelets is odd: the upper p44 band seems more phosphorylated than the lower p42 band, which is the dominant band in lysates from non-ASA treated platelets. Are the authors sure this is phospho-ERK since no molecular weights are provide on the blots?
Finally, it is an overstatement to say in the last sentence that daphnetin "may be used as a therapeutic drug to treat thromboembolic disorders". This statement should be removed or greatly changed to reflect the reality that much more preclinical data are needed to make this suggestion.
Author Response
We would like to thank the reviewer for the re-evaluation of the manuscript and helpful suggestions.
The responses to the specific comments for the reviewer are given below:
Response to Reviewer 2
Comment: These data imply a direct inhibition of cPLA2, which makes sense except for the fact that the antibody used to detect cPLA2 phosphorylation recognizes a known phosphorylation site for ERK (Ser505 on cPLA2). Are the authors suggesting that daphnetin inhibits ERK activity directly or does daphnetin bind cPLA2 and prevent phosphorylation at this site resulting in lower catalytic activity? This is a complicated signaling pathway with multiple positive feedback loops, so more discussion of how daphnetin may be regulating an ERK phosphorylation site on cPLA2 is warranted (preferably near Line 266 in the Discussion).
Response: We thank the reviewer for the comment. We agree with the reviewer that the cPLA2 antibody recognizes the Ser505 phosphorylation site for ERK indicating that ERK may influence cPLA2 activity in some way. However, we found that daphnetin inhibited 2-MeSADP-induced cPLA2 phosphorylation in aspirinated platelet while having no effect on ERK phosphorylation, suggesting that daphnetin does not directly target ERK to control cPLA2 activity. As per the reviewer’s suggestion, we have further discussed the possible mechanism of daphnetin in the regulation of platelet function in the revised version of the manuscript.
Comment: On a technical level, the new P-ERK1/2 blot from ASA-treated platelets is odd: the upper p44 band seems more phosphorylated than the lower p42 band, which is the dominant band in lysates from non-ASA treated platelets. Are the authors sure this is phospho-ERK since no molecular weights are provide on the blots?
Response: We thank the reviewer for the comment. We did not observe the p44 band in the 2-MeSADP-induced ERK phosphorylation in aspirinated platelets. In the previous revision, we simply displayed the p42 band but not p44 band of ERK phosphorylation. We apologize for this confusion. The phospho-ERK band has been accurately inserted in the revised version of the manuscript.
Comment: Finally, it is an overstatement to say in the last sentence that daphnetin "may be used as a therapeutic drug to treat thromboembolic disorders". This statement should be removed or greatly changed to reflect the reality that much more preclinical data are needed to make this suggestion.
Response: We thank the reviewer for the comment. We have removed the mentioned sentence in the revised version of the manuscript.
This manuscript is a resubmission of an earlier submission. The following is a list of the peer review reports and author responses from that submission.
Round 1
Reviewer 1 Report
The manuscript entitled " Anti-platelet Effect of Daphnetin is Regulated by ERK-mediated Thromboxane A2 Generation" in which the authors examined the impact of the coumarin by daphnetin on the regulation of platelet activation using murine platelets. They found that daphnetin has anti-platelet effect that is mediated by the inhibition of Thromboxane A2 generation through the regulation of ERK phosphorylation not AKT phosphorylation.
The data presented here supports the authors’ conclusions. The scientific narrative is well structured and flows naturally from one idea to the next. The findings are novel and important. Moreover, while this report provides a novel therapeutic compound for the treatment of thromboembolic disorders.
However, this paper suffers from some shortcomings that if modified would make the manuscript very suitable for publication in International Journal of Molecular Sciences.
Shortcomings:
1- Please add the species, sex, age and number of used mice for isolation of platelets in methods part?
2- Authors write in methods part “Levels of TXB2 were measured in duplicates using a TxB2 ELISA Kit” while the number of samples in each experiment and figure legend isn’t clear. Please write it in each figure legend.
3- Are the western blot data in figure 6 normalized to a housekeeping protein like beta actin or GAPDH?
4- In immunoblotting part in methods, please add the amount of protein which was separated on gel through SDS-PAGE.
5- How the relative phosphorylation of P-ERK1/2 and P-AKT was calculated? Is it calculated as a ratio from total ERK1/2 and total AKT respectively?
6- Why do the authors select these concentrations (25 µM, 50 µM, 100 µM) of Daphnetin?
7- Please add the word “unpaired” before t-test in line 284.
8- The authors write in their results “As shown in Figure 1, daphnetin significantly inhibited collagen-induced dense granule secretion in a dose-dependent manner, while only high concentration of daphnetin inhibited low concentration of collagen-induced platelet aggregation. The data suggest that daphnetin has a minor role in GPVI-mediated platelet response, unlike other coumarin derivatives”. How do the authors explain the low role of daphnetin in GPVI-mediated platelet response compared to other coumarin derivatives?
9- The MAPK signaling pathway, including ERK, JNK, and p38 MAPK, regulates platelet activation. Why do the authors examine the effect of daphnetin on ERK only and didn’t measure its effects on other MAPKs like p38 MAPK and JNK?

Author Response
Responses to Reviewer 1
Comment 1- Please add the species, sex, age and number of used mice for isolation of platelets in methods part?
Response: We thank the reviewer for the comment. We added the missing information in the manuscript. Please refer to the ‘Mouse Platelet Isolation’ and ‘Platelet Aggregation and Dense Granule Secretion’ part of the Materials and Methods section.
Comment 2- Authors write in methods part “Levels of TXB2 were measured in duplicates using a TxB2 ELISA Kit” while the number of samples in each experiment and figure legend isn’t clear. Please write it in each figure legend.
Response: We thank the reviewer for the comment. We revised the figure legend of each data in the manuscript. Please refer to the revised version of the manuscript.
Comment 3- Are the western blot data in figure 6 normalized to a housekeeping protein like beta actin or GAPDH?
Response: We thank the reviewer for the comment. We have used total ERK1/2 and total AKT as a loading control to normalize our western blot data in the manuscript.
Comment 4- In immunoblotting part in methods, please add the amount of protein which was separated on gel through SDS-PAGE.
Response: We thank the reviewer for the comment. In our western blot protocol, we fixed our platelet concentration to 2 ´ 108 cells/ml for each sample and prepared the lysate. Then, the lysate volume corresponding to the same number of platelets was loaded to SDS-PAGE gel and protein was separated. Total anti-ERK1/2 and total anti-AKT was used to analyze the loading control. As shown in the Figure 6, total ERK1/2 and total AKT band quantity were equal. As per the suggestion, we have revised the figure legend by adding ‘The lysate volume corresponding to the same number of platelets was loaded’. Please refer to the Figure 6 in the manuscript.
Comment 5- How the relative phosphorylation of P-ERK1/2 and P-AKT was calculated? Is it calculated as a ratio from total ERK1/2 and total AKT respectively?
Response: We thank the reviewer for the comment. Yes, we calculated the relative phosphorylation of P-ERK1/2 and P-AKT as a ratio from total ERK1/2 and total AKT, respectively.
Comment 6- Why do the authors select these concentrations (25 µM, 50 µM, 100 µM) of Daphnetin?
Response: We thank the reviewer for the comment. The IC50 value of daphnetin varies from 7.65 µM to 25.01 µM depending on different targets such as EFGR, PKA or PKC. Likewise, EC50 value also ranges from 31 µM to more than 650 µM. Therefore, the use of daphnetin concentration is highly variable depending on cell-type and time for treatment in the different in vitro experimental setting. Concentrations equivalent to 25-100 µM; 10-50 µM; 140-570 µM have been used in splenocytes proliferation assays, EGR receptor solution, and H. pylori clinical isolates experiments, respectively. As this is the first time we are using daphnetin to study its effect on platelet function, we chose to use the concentrations 25-100 µM to see the dose-dependent effect of daphnetin on platelet function.
Comment 7- Please add the word “unpaired” before t-test in line 284
Response: We thank the reviewer for the comment. We had performed ‘unpaired t-test’ to analyze our data and added the word “unpaired” before t-test in the revised version of the manuscript.
Comment 8- The authors write in their results “As shown in Figure 1, daphnetin significantly inhibited collagen-induced dense granule secretion in a dose-dependent manner, while only high concentration of daphnetin inhibited low concentration of collagen-induced platelet aggregation. The data suggest that daphnetin has a minor role in GPVI-mediated platelet response, unlike other coumarin derivatives”. How do the authors explain the low role of daphnetin in GPVI-mediated platelet response compared to other coumarin derivatives?
Response: We thank the reviewer for the comment. As mentioned in the discussion section, a very recent study evaluated the bioactivities of many daphnetin derivatives and suggested that daphnetin-based drugs have the potential to target GPCRs and can be used as novel therapeutic agents. TxA2 generation has a minor role in GPVI-induced platelet aggregation and that is the reason why daphnetin has a minor effect in collagen-induced platelet responses. Unlike daphnetin, the other coumarin derivatives in GPVI-mediated signaling have been known to act through PLCγ2 which is critical for GPVI signaling. In other words, if daphnetin inhibits PLCγ2 like other coumarin derivatives, we would not expect to see the aggregation in collagen-stimulated platelets in the presence of daphnetin. We also have added this discussion in the revised version of the manuscript. Please refer to Line 193-195, Page No. 10 in the manuscript.
Comment 9- The MAPK signaling pathway, including ERK, JNK, and p38 MAPK, regulates platelet activation. Why do the authors examine the effect of daphnetin on ERK only and didn’t measure its effects on other MAPKs like p38 MAPK and JNK?
Response: We thank and agree with the reviewer’s comment that the MAPK signaling pathway, including ERK, JNK, and p38 MAPK regulates platelet activation. There may be other targets of daphnetin by which it may affect the platelet function. However, ERK1/2 is the major MAPK that has been well established to play a major role in the regulation of TxA2 generation in platelets and thus critically regulates platelet function. The impact of p38 MAPK and JNK on platelet aggregation following different agonists still remains a matter of debate. Threshold concentration of thrombin- and CRP-induced platelet aggregation may be affected by p38 MAPK but high doses of CRP/thrombin stimulation bypasses p38 MAPK-dependent signaling and do not require the amplifying mechanism of p38 MAPK-driven TxA2 generation for sufficient platelet aggregation. Thus, in this study, we have focused on the effect of daphnetin on ERK1/2 in TxA2 generation. However, as the reviewer has suggested, we cannot rule out the minor contribution of other MAPKs, and other molecular target of daphnetin involved in platelet function needs further investigation.
Reviewer 2 Report
Chaudhary and collaborators sought to investigate the effects of Daphnetin, a coumarin derivative, on platelet aggregation and determine its mechanisms of action. The manuscript is interesting, but it has some major points to be clarified before it is considered for publication.
Major comments:
1) N numbers and repetitions have not been stated in the manuscript. Only figure 1 says 3 repetitions, but no n numbers, replicates or any other details. It is important to highlight that stats should not be performed with n numbers lower than 3
2) The authors stated that statistical comparisons were performed by t test, which is absolutely incorrect for the data they have (multiple groups). Thus, any assumptions from the manuscript cannot be considered based on the analysis performed
3) Details on antibodies should be given (i.e., concentrations, incubation periods, etc) as well as negative controls for the antibodies used
4) The anti-aggregation effects of Daphnetin should be confirmed in vivo.
Author Response
We would like to thank the reviewers and the editor-in-chief for the critical evaluation of the manuscript and helpful suggestions. We are thankful to the reviewers for finding our study novel and interesting. We believe that the issues raised are important, and we are thankful to the reviewers for their insightful comments and criticisms. We have clarified issues raised by reviewers and believe that addressing the criticisms has significantly improved the manuscript. The responses to the specific comments for all reviewers are given below:
Responses to Reviewer 2
Comment 1) N numbers and repetitions have not been stated in the manuscript. Only figure 1 says 3 repetitions, but no n numbers, replicates or any other details. It is important to highlight that stats should not be performed with n numbers lower than 3
Response: We thank the reviewer for the comment. We have performed each set of experiments for at least three independent times and statistical analysis has been performed with the data from those three independent sets of experiment. We have revised and mentioned the number of repetitions that was done during each experiment in the figure legend. Please refer to the revised version of the manuscript.
Comment 2) The authors stated that statistical comparisons were performed by t test, which is absolutely incorrect for the data they have (multiple groups). Thus, any assumptions from the manuscript cannot be considered based on the analysis performed
Response: We thank the reviewer for the comment. As you mentioned, the p-values estimated from t-tests should be adjusted, because we implemented multiple comparisons.
Using Bonferroni correction, we revised the p-values as followings.
Comment 3) Details on antibodies should be given (i.e., concentrations, incubation periods, etc) as well as negative controls for the antibodies used
Response: We thank the reviewer for the comment. As per the suggestion of the reviewer, we have included the details for the antibodies used in the revised version of the manuscript.
Comment 4) The anti-aggregation effects of Daphnetin should be confirmed in vivo.
Response: We thank the reviewer for the comment. The aim of this manuscript was to evaluate whether daphnetin has any effect on platelet functional responses and to explore the molecular mechanism that may be involved in the anti-platelet effect of daphnetin. Inhibition of ERK1/2 has been shown to significantly increase time to arterial occlusion suggesting a role of ERK1/2 in regulating thrombosis, but there have yet to be any studies confirming these results in genetic models. As daphnetin regulates the platelet function by inhibiting the TxA2 generation through the regulation of ERK1/2 phosphorylation, we expect to see the similar phenotype observed by inhibiting ERK1/2 in vivo. We will further fully dissect the effect of daphnetin on hemostasis and thrombosis in the future.
Round 2
Reviewer 2 Report
The authors have partially replied the comments made. They need to make clear in every legend to figure, the number of replicates and the number of experiments performed. In data analysis, they continued to state they performed t-test, in sets of data which all require anova followed by an appropriate post-hoc test.
They were also requested to demonstrate the effects of daphnetin in vivo. This point also remains not aswered as they did not perfom the experiment requested, nor discussed the importance of such analysis to further validate their hypothesis.
Author Response
We are thankful to the reviewers for their insightful comments and criticisms. We have further clarified issues raised by reviewers. The responses to the specific comments for all reviewers are given below:
Comment 1) They need to make clear in every legend to figure, the number of replicates and the number of experiments performed. In data analysis, they continued to state they performed t-test, in sets of data which all require anova followed by an appropriate post-hoc test.
Response: We thank the reviewer for the comment. We have revised as per the reviewer’s comment. We have mentioned the number of experiments performed and number of samples for each experiment in every figure legend. Further, we apologize for not performing the ANOVA test from the first. In the last revision, we had consulted the statistician of our department and corrected the analysis of our data using Bonferroni correction. However, this time we have performed ANOVA followed by post-hoc test and revised the figure accordingly. Please refer to the revised manuscript.
Comment 2) They were also requested to demonstrate the effects of daphnetin in vivo. This point also remains not answered as they did not perform the experiment requested, nor discussed the importance of such analysis to further validate their hypothesis.
Response: We thank the reviewer for the comment. We agree with the reviewer that in vivo experiment would clearly elucidate the role of daphnetin in platelet function and validate our hypothesis. Although conducting the in vivo experiment is the limitation of our study, we have discussed the importance of anti-aggregation effects of daphnetin in vivo in the discussion section to further dissect the physiological effect of daphnetin in the manuscript.
Round 3
Reviewer 2 Report
The authors have now provided additional information on n numbers. Did they perform power analysis to determine the ideal sample size for their experiments? Do the data fits normal distribution?
The authors still failed to convince on the absence of in vivo data. They must to provide further discussion on this matter.
Author Response
We would like to thank the reviewer for the re-evaluation of the manuscript and helpful suggestions. We are thankful to the reviewer for the insightful comments and criticisms. We have further clarified the issues raised by the reviewer. The responses to the specific comments are given below:
Responses to Reviewer 2
Comment 1) The authors have now provided additional information on n numbers. Did they perform power analysis to determine the ideal sample size for their experiments? Do the data fits normal distribution?
Response: We thank the reviewer for the comment. We have used a minimum sample size to statistically analyze our data. This sample size has been widely used to perform a variety of experiments and validate the data statistically. For our experiments, we did not perform pre-power analysis to determine the ideal sample size, however, as per the reviewer’s suggestion, we performed a post hoc test after ANOVA to establish our statistical analysis in our last revision. In addition, as shown in the figures, even with the minimum sample size (low power), data were statistically significant suggesting that the sample size used in our experiment is ideal and believe that the data can be accepted. Now, we also checked the normality test by using the Shapiro-Wilk test and found that our data were normally distributed (p > 0.05).
Comment 2) The authors still failed to convince on the absence of in vivo data. They must to provide further discussion on this matter.
Response: We thank the reviewer for the comment. We apologize for not being able to follow the reviewer’s comment absolutely. As we mentioned earlier, conducting the in vivo experiment is the limitation of our study currently. Therefore, we have added some additional discussion on the importance of conducting in vivo effect of daphnetin in platelet function, which is important to understand whether daphnetin can be used as an alternative therapeutic source of TxA2 inhibitor to treat various cardiovascular diseases. Please refer to the revised version of the manuscript.
Round 4
Reviewer 2 Report
Unfortunately, although interesting, due to the lack of in vivo data supporting the in vitro evidence of the study and the lack of plausible justification for n numbers used, the manuscript publication cannot be yet endorsed.